# Features for medically serious suicide attempters who do not have a strong intent to die: a cross-sectional study in rural China

Long Sun,[1,2] Jie Zhang,[1,3] Dorian A Lamis[4]

[1]Center for Suicide Prevention Research, School of Public Health, Shandong University, Jinan, China
[2]Key Laboratory of Health Economics and Policy Research, National Health Commission of China, Jinan, Shandong, China
[3]Department of Sociology, State University of New York Buffalo State, Buffalo, New York, USA
[4]Department of Psychiatry and Behavioral Sciences, Emory University School of Medicine, Atlanta, Georgia, USA

**Correspondence to**
Professor Jie Zhang;
zhangj@buffalostate.edu

## ABSTRACT

**Objective** Previous studies have implied that there were many Chinese suicide attempters who did not want to die by suicide. In the current study, we explored the factors which were associated with low levels of suicide intent. We also examined features for medically serious suicide attempters who do not have a strong intent to die in rural china.

**Design** Cross-sectional study.

**Setting** The interviews occurred between May 2012 and July 2013 in 13 rural counties in Shandong and Hunan provinces, China.

**Participants** Subjects were 791 medically serious suicide attempters whose injury and wounds were so serious they required hospitalisation or immediate medical care.

**Results** The results supported that less years of education (β=−0.11, p=0.037), religious beliefs (β=1.20, p=0.005), living alone (β=1.92, p=0.017), negative life events (β=0.29, p=0.003), low levels of impulsivity (β=−0.10, p=0.013) and mental disorders (β=2.82, p<0.001) were associated with higher levels of suicide intent.

**Conclusion** Results imply that there are some medically serious suicide attempters with a higher education and/or exhibit impulsivity who do not want to die by suicide. These findings can inform practice to prevent suicide in rural China.

## BACKGROUND

The WHO estimated that there were about 804 000 suicide deaths worldwide in 2012 which equated to one person dying by suicide every 40 s.[1] With regard to suicide attempts, some have suggested that the number was about 20 times higher than suicide death,[2] and nearly 50% of attempts required emergency medical treatment.[3 4] China has one of the higher suicide rates in the world.[5] Although the rates have decreased in recent years, suicide attempts and deaths are also important social and public health issues in China.[6]

In the last decades, several studies have explored the patterns of suicide and found differences between China and other

### Strengths and limitations of this study

► This study is based on a large sample of suicide attempters in rural China (n=791).
► Medically serious suicide attempters were consecutively recruited in selected emergency rooms which ensured the validity of the sample.
► This is one of few studies examining the factors associated with intent among medically serious suicide attempters in rural China.
► As this is a cross-sectional study, we cannot infer causal relationships among study variables.
► All of the attempters were interviewed following hospital discharge, so recall bias is a possibility.

countries.[7] Many of these investigations imply that there are some suicide attempters who do not intend to die by suicide and may only instrumentally use for some other aims, such as getting attention from their family members, proving their viewpoint or behaviour.

First, there is a difference in percentages of mental disorders in suicide across countries. In Western countries, it has been estimated that 90% of individuals who die by suicide have a diagnosable mental disorder.[8] However, this percentage was lower in comparison to Asian countries. For example, they were only about 40%–70% in mainland China, 16.2% in India and 12.0% in Malaysia.[9 10] Although mental disorders remain an important risk factor for suicidal behaviour, other factors may play roles in Asian suicides. Thus, exploring the characteristics of suicide attempters in China not only may help us to understand the differences between China and Western countries, but also can provide important information about other Asian countries.

Second, impulsivity has been identified as another risk factor for suicide attempts both in China and Western countries.[11 12] However, previous studies have found that

BMJ

approximately 50% of suicide attempters in China could be categorised as impulsive.[13] It further implies that there is a large group of Chinese suicide attempters who do not really want to die by suicide, and their suicidal behaviours may be impulsive.

As mentioned above, many findings about Chinese suicide suggest that there are some suicidal individuals who do not really want to die by suicide. Suicide intent is defined as the level of intent to die by suicide,[14] and many studies show that it is significantly associated with suicidal behaviour.[15] Investigating suicide intent is helpful for us to better understand the suicide attempters who do not really want to die.

In recent years, many studies identified factors associated with suicide intent. In Western countries, researchers have found that older adults,[16] rural residence,[17] mental health problems[18] and hopelessness[19] were associated with suicide intent. In China, there were also some studies which supported that older age, higher level of education, living alone and suicide communication were correlated with higher level of suicide intent among individuals who died by suicide.[20 21] However, we have little knowledge about the suicide intent among suicide attempters who do not really want to die by suicide in China.

Thus, in the current study, we explored factors associated with low level of suicide intent. It was helpful for us to understand the features for medically serious suicide attempters who do not have a strong intent to die in rural China which may inform intervention and prevention strategies for at-risk individuals in rural China.

## METHODS
### Study sample and the design
In the current study, all of attempters were chosen from two provinces in China: Shandong and Hunan. Shandong is located in the north of China, and is a province with economic prosperity in both industry and agriculture. Hunan is located in the south of China, and is a province with economic prosperity in agriculture. In the two provinces, 13 rural counties were randomly selected.

In each of the rural county from May 2012 to July 2013, all hospital emergency departments were instructed to notify the research teams in each province when suicide attempts occurred. We consecutively recruited the attempters who were aged 15–54 years in either rural region. The enrolment of patients followed the definition of suicide attempt and deemed medically serious. In this study, suicide attempt was defined as someone who attempted suicide and wanted to die, but did not,[22 23] and medically serious suicide attempters included those survivors whose injuries and wounds were so serious as to require hospitalisation or immediate medical care.

All of the interviewers received training about the study and were master or PhD level students in the medical school. The main aims of this training were to provide the interviewers with sufficient information about the study and questionnaire.

### Interviewing procedures
All of the attempters were interviewed following hospital discharge. In order to successfully contact the attempters, all of them were first approached in-person by the local health agency and/or village administration. On their agreement of written informed consent, the interview time was scheduled approximately 2 months after suicide incident. Each attempter was interviewed separately by one trained interviewer in private at the village medical room or their home. For those participants who were too weak to talk, family members could assist in the interview by answering some of the questions on the protocol. The average time for each interview was 1.5 hours.

### Patient and public involvement statement
The suicide attempters were first involved in the process of data collection. The aims of this study and outcome measures were informed via the interviewers. There are no plans to disseminate the qualitative study results to subjects or the relevant patient community.

### Measures
#### Suicide intent
Beck's Suicidal Intent Scale (SIS) was used to measure the degree of suicide intent for the attempters.[24] The SIS assesses attempters' precautions, planning, communication and expectations about the suicidal behaviour. There are 15 items on this scale, and each item is scored from 0 to 2. The psychometric properties of the English version of SIS has been evaluated among suicide attempters and decedents.[25] The Chinese version of SIS also demonstrated sound reliability and validity which was shown in a previous study.[26]

#### Social-demographic variables
*Age* which ranged from 15 to 54 years was calculated to the time when the suicide occurred. *Gender* was measured by male or female. *Education years* were evaluated by the number of years in which the attempters completed in school. *Married status* was dichotomised as 'never married' and 'ever married' with the latter including those who were divorced, separated or widowed. *Occupation* was assessed by peasant, businessman, public service staff, student, factory worker, rural doctor, teacher, housewife, unemployed and others. As most attempters were peasants, we recoded the variable into peasants and others. *Religious belief* was measured by what religion the attempters believed in, and the choices were Taoism, Muslim, Christianity, Buddhism, others and no religion. As there were few people who had a religious belief, the religious belief was recoded into 'yes' or 'no'.

#### Living alone
Living alone was assessed by a question that 'Do you live with others?' with response options being 'yes' or 'no'. Participants who did not live with others were deemed as living alone. The same evaluation method was used in our previous suicide studies.[27]

## Physical disease

Physical disease was assessed by one question: 'Have you been diagnosed with a chronic disease now?' with response options including 'yes' or 'no'.

## Pesticide at home

Pesticide availability at home was assessed with a single item which asked the participants if any type of farming chemicals were stored at home. The effect of pesticide on suicide has been shown in previous Chinese studies.[28] The response options consisted of 'yes' or 'no'.

## Family suicide history

Family suicide history was measured by a question: 'Do your family members conduct suicide behaviour before?' The answer also could be chosen from 'yes' or 'no'.

## Negative life events

Negative life events (NLEs) were determined by the revised version of Interview for Recent Life Events (IRLE).[29] The IRLE is a 64-item scale which measures life events occurring in the past 12 months. We also asked another question in case there were other life events which were not asked in the 64 items. The attempters could also answer if each event was perceived as positive or negative. In this study, we only used the number of NLEs. The Chinese version of IRLE has been used in previous suicide studies.[30]

## Impulsivity

The 23-item Dickman Impulsivity Inventory (DII) was used to evaluate the level of impulsivity which was developed and validated in English.[31] Each item included a response option of yes (1) or no (0). The sum score for all items was used in the data analysis, and the higher score represented a higher level of impulsivity. The Chinese version of the DII has been tested and demonstrated sound reliability and validity.[32]

## Coping skills

Coping Responses Inventory (CRI) was used to assess the attempters' coping skills in this study.[33] It asked the participants to evaluate the frequency (0=never, 1=occasionally, 2=sometimes, 3=often) of engaging in 48 separate coping activities. Sample questions on the CRI included 'talk with your spouse or other relative about the problem' and 'think about how this event could change your life in a positive way'. The Chinese version of CRI had been used in previous suicide studies.[34]

## Mental disorder

We used the Chinese version of the Structured Clinical Interview for the Diagnostic and Statistical Manual of Mental Disorders (SCID)[35] to determine diagnoses for suicide attempters. Diagnoses were made by the psychiatrists with the written information obtained by the trained interviewers for each suicide attempt. There was one psychiatrist to make the diagnosis in each province. The Chinese version of the SCID was provided by the Department of Psychiatry of Kaohsiung Medical College in Taiwan,[36] and permission to use the work had been obtained. It also had been used in Chinese populations in many areas, including Taiwan, Hong Kong, Macau and mainland China for the past few decades.[37] A total of 27 axis I mental diseases were detected by the SCID, and we used the dichotomous diagnosis for each of them with yes and no.

## Statistical methods

IBM SPSS Statistics V.24.0 (Web Edition) was used for the data analysis. Student's t-tests or bivariable correlation analysis were used to compare the differences in suicide intent among categorical and continuous variables. Stepwise linear regression was conducted to examine the factors related to suicide intent. All of the tests were two-tailed, and a p value of <0.05 was considered statistically significant.

## RESULTS

In the current study, 791 suicide attempters were successfully interviewed. Table 1 describes the sample distribution regarding social and psychological characteristics. The average age and education years were 31.63 and 6.90, respectively. Among these attempters, there were more females (63.0%) and peasants (53.4%). The majority of attempters were ever married (83.3%), did not believe in religion (81.3%), did not suffer from a physical disease (83.2%), stored pesticides at home (60.4%), did not live alone (95.6%), did not have a family history of suicide (92.9%) or were diagnosed with a mental disorder (80.9%). The mean for NLEs, impulsivity and coping skills were 1.83, 9.89 and 33.13, respectively.

We also examined the differences in suicide intent among these social and psychological characteristics. Results demonstrated that age (r=0.071, p=0.045), education years (r=−0.076, p=0.032), religious belief (t=3.340, p=0.001), living alone (t=2.315, p=0.021), physical disease (t=2.416, p=0.016), NLEs (r=0.148, p=0.001), impulsivity (r=−0.084, p=0.019) and mental disorder (t=7.393, p=0.001) were associated with suicide intent (see table 2).

Results of stepwise linear regression examining factors associated with suicide intent are presented in table 3. We found that less education years (β=−0.11, p=0.037), religious belief (β=1.20, p=0.005), living alone (β=1.92, p=0.017), NLEs (β=0.29, p=0.003), low level of impulsivity (β=−0.10, p=0.013), mental disorder (β=2.82, p=0.001) were associated with higher levels of suicide intent.

## DISCUSSION

The present study focused on suicide intent among medically serious suicide attempters in rural China. The primary purpose was to explore the features for medically serious suicide attempters who did not have a strong intent to die. Results indicated that the attempters with a strong intent to die were associated with religious belief,

**Table 1** Description of the social and psychological characteristics among Chinese rural medically serious suicide attempters (n=791)

| Variables | Mean±SD/n (%) |
|---|---|
| Age | 31.63±8.00 |
| Gender | |
| Male | 293 (37.0) |
| Female | 498 (63.0) |
| Education years | 6.90±3.26 |
| Married status | |
| Never married | 132 (16.7) |
| Ever Married | 659 (83.3) |
| Occupation | |
| Peasant | 422 (53.4) |
| Others | 369 (46.6) |
| Religious belief | |
| Yes | 148 (18.7) |
| No | 643 (81.3) |
| Living alone | |
| Yes | 35 (4.4) |
| No | 756 (95.6) |
| Physical disease | |
| Yes | 133 (16.8) |
| No | 658 (83.2) |
| Pesticide at home | |
| Yes | 478 (60.4) |
| No | 313 (39.6) |
| Family suicide history | |
| Yes | 56 (7.1) |
| No | 735 (92.9) |
| Negative life events | 1.83±1.77 |
| Impulsivity | 9.89±4.08 |
| Coping skill | 33.13±10.16 |
| Mental disorder | |
| Yes | 151 (19.1) |
| No | 640 (80.9) |

**Table 2** Comparing the suicide intent among the social and psychological characteristics (n=791)

| Variables | Suicide intent (mean±SD) | t/r | P values |
|---|---|---|---|
| Age | – | 0.071 | 0.045 |
| Gender | | 0.342 | 0.559 |
| Male | 9.37±4.96 | | |
| Female | 9.86±4.84 | | |
| Education years | – | −0.076 | 0.032 |
| Married status | | 1.179 | 0.239 |
| Never married | 9.22±4.88 | | |
| Ever married | 9.77±4.89 | | |
| Occupation | | 1.811 | 0.071 |
| Peasant | 9.97±4.92 | | |
| Others | 9.34±4.84 | | |
| Religious belief | | 3.430 | 0.001 |
| Yes | 10.91±4.82 | | |
| No | 9.39±4.87 | | |
| Living alone | | 2.315 | 0.021 |
| Yes | 11.54±5.52 | | |
| No | 9.59±4.85 | | |
| Physical disease | | 2.416 | 0.016 |
| Yes | 10.61±4.93 | | |
| No | 9.49±4.86 | | |
| Pesticide at home | | −0.980 | 0.327 |
| Yes | 9.54±4.66 | | |
| No | 9.89±5.23 | | |
| Family suicide history | | 2.104 | 0.036 |
| Yes | 11.00±5.18 | | |
| No | 9.58±4.86 | | |
| Negative life events | – | 0.148 | 0.000 |
| Impulsivity | – | −0.084 | 0.019 |
| Coping skill | – | −0.067 | 0.061 |
| Mental disorder | | 7.393 | 0.000 |
| Yes | 12.24±4.77 | | |
| No | 9.07±4.72 | | |
| All | 9.68±4.89 | – | – |

living alone, NLEs and mental disorder; whereas, the low intent suicide attempters had less years of education and more impulsivity.

We found that education years were negatively associated with suicide intent. It means that the attempters with higher education have lower intention to die by suicide behaviour. Many studies have identified that higher education is a protective factor for suicide behaviour worldwide.[38 39] In China, previous studies also support this relationship between education and suicide intent among rural suicides.[40]

Religious belief is another factor associated with strong suicide intent. As there are few people who have a religious belief in China,[41] many people see religious belief as a deviant behaviour. Suicide which also can be seen as a deviant behaviour in the society may be correlated between them. Besides, many Chinese rural residents believe in a religion after they suffer difficulties or misfortunes. Suffering difficulties or misfortunes, as an important risk factor for suicide,[42] may promote the intent to die by suicide in rural China. This is why attempters with religious belief have strong intent to die by suicide. It is different from the findings in Western countries.[43]

**Table 3** Linear regression about the social and psychological characteristics associated with suicide intent (n=791)

| Variables | β | 95% CI | P values |
|---|---|---|---|
| Education years | −0.11 | −0.21 to 0.01 | 0.037 |
| Religious belief | 1.20 | 0.36 to 2.04 | 0.005 |
| Living alone | 1.92 | 0.34 to 3.51 | 0.017 |
| Negative life events | 0.29 | 0.10 to 0.48 | 0.003 |
| Impulsivity | −0.10 | −0.18 to 0.02 | 0.013 |
| Mental disorder | 2.82 | 1.98 to 3.66 | 0.000 |
| Constant | 10.04 | 8.91 to 11.16 | 0.000 |

Adjust $R^2$=0.097.
Stepwise linear regression was used in this regression.

We also found living alone and NLEs were associated with suicide intent among attempters. Both of them have been shown to be risk factors for suicide behaviour in previous studies worldwide.[44 45] People living alone often find it difficult to communicate with others which is a risk factor for suicide intent.[46] Individuals who encounter NLEs often experience psychological problems which contribute to suicidal behaviour. In the same way, people who live alone and suffer NLEs may have a strong intent to die by suicide.

Impulsivity has been identified as an important risk factor for suicide in many studies.[47 48] As we discussed above, many Chinese suicide attempters can be categorised as impulsive.[13] One of the possible reasons may be that borderline personality disorder is associated with impulsivity and self-harm behaviour, but was not diagnosed in the current study.[49] Previous studies have also suggested that impulsivity contributed to suicide death in rural China.[21] It implies that some attempters do not want to die by suicide which is consistent with our assumption.

The present study also identified mental disorders as an important risk factor for suicidal behaviour.[50–52] An individual diagnosed with a mental disorder suffers from psychological symptoms which contribute to an increased risk for suicide. Thus, attempters with mental disorders may want to die more than those who do not have any mental disorder.

In the current study, we did not find gender differences in suicide intent among suicide attempters. Previous studies demonstrated that male attempters tend to choose violent and lethal methods.[53] It was easy for us to conclude that males may experience higher levels of suicide intent. However, the choice of the suicide means may be caused by the higher level of violence for men compared with women, and we cannot conclude that men may have a higher intent to die.[21] In the current study, we also examined the relationship between physical disease, family suicide history and suicide intent. Both of them were not associated with suicide intent. Physical disease was associated with suicidal behaviour which was supported by previous studies.[54] However, physical disease did not have a stronger association with those identified with a stronger intent to die compared with those who did not have a strong intent to die. Previous studies have found that physical disease may lead to a mental disorder, increasing the likelihood for suicidal behaviour.[55] Although some studies found family suicide history was associated with suicidal behaviour,[56] we did not find any evidence showing this relation.

There were several limitations to our study which should be considered when interpreting these findings. First, as this is a cross-sectional study, we cannot infer any causal relationship for the study variables. Second, all of the attempters were interviewed following discharge from hospitals, so recall bias is a potential confounder. Third, the participants in this study were all medically serious suicide attempters, and the results may not be consistent with other types of suicide attempt. Finally, lethality, which is an important factor associated with suicide intent, was not investigated in our study.

## CONCLUSION

Despite these limitations, this study contributes to our understanding of Chinese suicidal behaviour. Our results support that there are some medically serious suicide attempters with higher education and impulsivity who do not really want to die by suicide. These findings can inform suicide assessment and intervention to prevent suicide in China.[57]

**Acknowledgements** We would like to thank the participants in this study, and we also want to thank the research teams in China for their field work in the data collection.

**Contributors** LS analysed the data and wrote the manuscript, JZ designed the study and reviewed the paper and DAL reviewed and edited the manuscript.

**Funding** The research was supported by the US National Institute of Mental Health (R01 MH068560) and National Natural Science Foundation of China (71603149).

**Competing interests** None declared.

**Patient consent** Obtained.

**Ethics approval** The IRB approvals from both the Chinese institutions (Shandong University and Central North University) and the US-based university (State University of New York, Buffalo State) where the principal investigator is affiliated ensured the human subjects protection and the ethical methodology regulated by NIMH which funded the project.

**Provenance and peer review** Not commissioned; externally peer reviewed.

**Data sharing statement** The data and materials used in this study are available from the first author (LS) and corresponding author (JZ) on reasonable request.

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
