## [Reviewer comments · BMJ Open]

ARTICLE DETAILS

TITLE (PROVISIONAL)	Features for Medically Serious Suicide Attempters Who Do Not Have a Strong Intent to Die: A Cross-sectional Study in Rural China
AUTHORS	Sun, Long; Zhang, Jie; Lamis, Dorian

VERSION 1 – REVIEW

REVIEWER	Roger Ho Department of Psychological Medicine National University of Singapore
REVIEW RETURNED	03-Jun-2018

GENERAL COMMENTS	Thank you for inviting me to review the paper on “Features for Medically Serious Suicide Attempters Who Do Not Have a Strong Intent to Die in rural China”. This is an important study which deserves to be published in the BMJ Open. I have the following recommendations to improve the quality of the paper before publication. 1. The authors need to get language editing service or include a co-author who has good command of academic English to edit the language of this paper. The following examples highlight the language problems: “In these findings, many of them imply that there are some suicides who do not intent to die by suicide, and they may only instrumentally use for certain gains.” We cannot call those people who attempted suicide as “some suicides”. The term, “certain gains” is vague. “Their suicide behaviour was promoted by impulsivity.” The term “promoted” has a positive connotation. A sales person promotes a product to his or her customers. As suicide behaviour leads to negative consequences, it is inappropriate to use the term “promote”. “In the current study, all of attempters were chose from two provinces in China,” It should be “chosen” “All the attempters were interviewed when they had leaved hospitals because of their weakness in the hospitals.” What does “weakness” mean? “In Table 1, we described the sample distribution about the social and psychological characteristics among Chinese rural medically serious suicide attempters.” There are too many adjectives: rural, medically and serious.
---

	There are many examples in this paper which require language editing. 2. In the introduction, the authors mainly mentioned about the differences in suicide between western and eastern countries. It is important to highlight findings among Asians or Chinese living in a society with western values and lifestyles (e.g. Singapore). I recommend the authors to discuss the following findings in line 25-40 of Pg 5. In Asian society that is influenced by western culture and values (e.g. Singapore), studies have found that religion (Mak et al 2015) and resolution of conflict (Choo et al 2017a) are protective factors against suicide. Usage of alcohol is an important precipitating factor (Choo et al 2014) while past psychiatric history is an important predisposing factor for suicide (Ho RC et al 2014). Thematic analysis showed that relationship issues, financial strain, socio-legal-academic and environmental stress, physical and mental illness as well as pain are dynamic risk factors for suicide in Asians (Choo et al 2018). In comparison to other Asian ethnic groups, around 33.4% of Chinese suicide attempters had mental illness. This proportion was higher in comparison with 16.2% of Indians and 12% of Malays (Choo et al 2017b). For protective factors, 60% of Malay attempters reported presence of religious beliefs, this proportion was higher in comparison with 44.8% of Indians and 26.6% of Chinese (Choo et al 2017b). Please include the following important references: Mak KK et al Ethnic differences in suicide behavior in Singapore. Transcult Psychiatry. 2015 Feb;52(1):3-17. doi: 10.1177/1363461514543545. PMID:25062745 Choo CC et al (2017a) What predicts medical lethality of suicide attempts in Asian youths? Asian J Psychiatr. 2017 Oct;29:136-141. doi: 10.1016/j.ajp.2017.05.008. Epub 2017 May 15. PMID: 29061412 Choo C et al (2014) Cluster analysis reveals risk factors for repeated suicide attempts in a multi-ethnic Asian population. Asian J Psychiatr. 2014 Apr;8:38-42. PMID:24655624 Ho RC et al (2014). Elderly suicide with and without a history of suicidal behavior: implications for suicide prevention and management. Arch Suicide Res. 2014;18(4):363-75. PMID:24828390 Choo CC et al (2018) Thematic Analysis of Medical Notes Offers Preliminary Insight into Precipitants for Asian Suicide Attempters: An Exploratory Study. Int J Environ Res Public Health. 2018 Apr 20;15(4). PMID: 29677141 Choo CC et al (2017b) Does ethnicity matter in risk and protective factors for suicide attempts and suicide lethality? PLoS One. 2017 Apr 20;12(4):e0175752. doi: 10.1371/journal.pone.0175752. eCollection 2017. PMID:28426687 3. On Pg 11, line 29, the authors stated that low level of impulsivity is associated with high level of suicide intent. The label of low level
--	---

of impulsivity sounds contradictory. I suggest the authors to clarify the description of impulsivity.

4. On Pg 12, line 48, the authors stated that, "as we introduced above, many Chinese suicide attempters can be diagnosed with impulsivity suicide." First, impulsivity suicide is not a psychiatric diagnosis. Second, there is a possibility that the diagnosis of borderline personality disorder was missed as this condition is prevalence in women and characterised by impulsivity and recurrent self-harm behaviour. The authors should discuss the following finding:

In Asian patients, symptoms of borderline personality disorder were associated with risk of repeated overdoses with benzodiazepines and paracetamol (Choo et al 2014) and this condition is underdiagnosed.

Reference:

Choo C et al (2014) Cluster analysis reveals risk factors for repeated suicide attempts in a multi-ethnic Asian population. *Asian J Psychiatr.* 2014 Apr;8:38-42. PMID:24655624

5. On Pg 13, line 9, the authors mentioned about mental disorders without specifying the type of mental disorders. As more than half of suicide attempters were women and the mean age was around 31 years. The authors should discuss postnatal depression. There is a good BMJ Open article reviewing smartphone applications to engage women with postnatal depression. Smartphone applications may have the potential to engage women in rural area. Please refer to this study.

Zhang MW, et al (2017). Current status of postnatal depression smartphone applications available on application stores: an information quality analysis. *BMJ Open.* 2017 Nov 14;7(11):e015655. PMID:29138195

6. Under discussion, the authors need to comment why there was no gender difference. This finding is different from other Asian suicide attempters living in societies with western influence. In other Asian studies, more males than females made attempts with high perceived lethality and high medical lethality (Choo et al 2017). The authors should discuss why the rural Chinese are different from other Asians.

Reference:

Choo CC et al (2017) Prediction of Lethality in Suicide Attempts: Gender Matters. *Omega (Westport).* 2017 Jan 1:30222817725182. PMID:28828921

7. Under discussion, I suggest the authors to develop a brief suicide assessment based on the factors identified and previous model of prediction. Please add the following statement in the discussion:

Based on our research findings, brief suicide risk assessment that include education level, negative life events and religious belief will achieve stronger predictive ability and reliability in suicide prediction based on the Affect-Behavior-Cognition model (Harris et al 2015).

	Please include the following references: Harris KM et al (2015) The ABC's of Suicide Risk Assessment: Applying a Tripartite Approach to Individual Evaluations. PLoS One. 2015 Jun 1;10(6):e0127442. PMID:26030590 8. I recommend the authors to discuss their findings with reference to the following important BMJ open articles in the discussion. Xu H, et al (2016) A cross-sectional study on risk factors and their interactions with suicidal ideation among the elderly in rural communities of Hunan, China. BMJ Open. 2016 Apr 15;6(4):e010914. PMID: 27084285 Sun J et al (2015) Incidence and fatality of serious suicide attempts in a predominantly rural population in Shandong, China: a public health surveillance study. BMJ Open. 2015 Feb 11;5(2):e006762 PMID:25673439 Zhang M et al (2013).Pesticide poisoning in Zhejiang, China: a retrospective analysis of adult cases registration by occupational disease surveillance and reporting systems from 2006 to 2010. BMJ Open. 2013 Nov 21;3(11): e003510. PMID: 24270833
--	--

REVIEWER	Maurizio Pompili Sapienza University of Rome
REVIEW RETURNED	14-Jul-2018

GENERAL COMMENTS	The authors reported original research report pointing to features for medically serious suicide attempters whose lethality seem limited. The paper is of interest for the journal. I have however some comments to report. First, aims and hypothesis should be better described. Second, authors should clearly describe what a suicide attempt it, and possibly refer to the revision of nomenclature in suicidology such as Silverman et al, 2007 (two papers). Third, authors may wish to discuss the concept of lethality as the probability to die by suicide and refer to the thought of Shneidman. Authors may also need to add some details referred to the enrollment of patients, who actually assessed them, if more than a psychiatrist was involved. Finally, I would point to the prevention of such act and discuss the following paper: The communication of suicidal intentions: a meta-analysis. Psychol Med. 2016
---

VERSION 1 – AUTHOR RESPONSE

Reviewer(s)' Comments to Author:

Reviewer: 1

Reviewer Name: Roger Ho

Institution and Country: Department of Psychological Medicine, National University of Singapore

Please state any competing interests or state 'None declared': Nil

-Thank you for inviting me to review the paper on "Features for Medically Serious Suicide Attempters Who Do Not Have a Strong Intent to Die in rural China". This is an important study which deserves to be published in the BMJ Open. I have the following recommendations to improve the quality of the

paper before publication.

[Response] Thank you so much for your nice comments. We have tried our best to revise this manuscript, and we wish all of our responses can meet kind consideration. Following is our responses to your comments.

-1. The authors need to get language editing service or include a co-author who has good command of academic English to edit the language of this paper. The following examples highlight the language problems:

[Response] Thank you for your suggestion. The editor also listed the language problem. Dr. Dorian Lamis, a native speaker, has helped us to revise the manuscript. All of the revisions have been marked in the manuscript. Please check the revised version.

-“In these findings, many of them imply that there are some suicides who do not intent to die by suicide, and they may only instrumentally use for certain gains.” We cannot call those people who attempted suicide as “some suicides”. The term, “certain gains” is vague.

[Response] Thank you so much for your earnest on our manuscript. For the first term, some suicides have been revised into “some suicide attempters.” I do agree with your comment on the term of certain gains. Actually, it is really hard for us to find a word to explain this. As you may know, in rural China, there are many kinds of aims for suicide behaviors. For example, somebody want to get attention from their family members, somebody want to prove their viewpoint or behavior. There are also many other aims for the suicide behavior. Some examples can be found in a Science article 2. This is the real meaning which we want to say. As a response, we have revised it into “some other aims, getting attention from their family members, proving their viewpoint or behavior.” We wish this term can meet our thinking about it.

-“Their suicide behaviour was promoted by impulsivity.” The term “promoted” has a positive connotation. A sales person promotes a product to his or her customers. As suicide behaviour leads to negative consequences, it is inappropriate to use the term “promote”.

[Response] Thank you so much for your comments. After this, we get more knowledge about the word “promote.” We are sorry about our poor English. We have revised it into “and their suicidal behaviors may be impulsive.”

-“In the current study, all of attempters were chose from two provinces in China,” It should be “chosen”

[Response] It is really our careless. We are sorry about it. We have revised this word into “chosen.” Thank you so much.

-“All the attempters were interviewed when they had leaved hospitals because of their weakness in the hospitals.” What does “weakness” mean?

[Response] We are sorry about our misunderstanding about it. In this study, all of the attempters were interviewed at their home or village. Actually, we can get more accurate data if we can interview them in a short time after the suicide behavior. However, they are too weak in the hospitals, because all of the subjects in our study required medical treatment. Considering the safety and the ethical problem, we interviewed them when they had leaved hospital. As a response, we have deleted the term that “because of their weakness in the hospitals.”

-“In Table 1, we described the sample distribution about the social and psychological characteristics among Chinese rural medically serious suicide attempters.” There are too many adjectives: rural, medically and serious.

[Response] Thank you for your suggestion. We have deleted the sentence “among Chinese rural medically serious suicide attempters.” It is the same meaning with the word “sample.” The detailed information about the sample has been introduced in the method section.

-There are many examples in this paper which require language editing.

[Response] It is similar response to your first comment. We have tried our best to revise the language problems in this manuscript. All of the revisions have been marked in the revised manuscript. Thank you again for your earnest on our manuscript, all of these are so helpful for our revising.

-2. In the introduction, the authors mainly mentioned about the differences in suicide between western and eastern countries. It is important to highlight findings among Asians or Chinese living in a society

with western values and lifestyles (e.g. Singapore). I recommend the authors to discuss the following findings in line 25-40 of Pg 5.

In Asian society that is influenced by western culture and values (e.g. Singapore), studies have found that religion (Mak et al 2015) and resolution of conflict (Choo et al 2017a) are protective factors against suicide. Usage of alcohol is an important precipitating factor (Choo et al 2014) while past psychiatric history is an important predisposing factor for suicide (Ho RC et al 2014). Thematic analysis showed that relationship issues, financial strain, socio-legal-academic and environmental stress, physical and mental illness as well as pain are dynamic risk factors for suicide in Asians (Choo et al 2018). In comparison to other Asian ethnic groups, around 33.4% of Chinese suicide attempters had mental illness. This proportion was higher in comparison with 16.2% of Indians and 12% of Malays (Choo et al 2017b). For protective factors, 60% of Malay attempters reported presence of religious beliefs, this proportion was higher in comparison with 44.8% of Indians and 26.6% of Chinese (Choo et al 2017b).

Please include the following important references:

Mak KK et al Ethnic differences in suicide behavior in Singapore. *Transcult Psychiatry*. 2015 Feb;52(1):3-17. doi: 10.1177/1363461514543545. PMID:25062745

Choo CC et al (2017a) What predicts medical lethality of suicide attempts in Asian youths? *Asian J Psychiatr*. 2017 Oct;29:136-141. doi: 10.1016/j.ajp.2017.05.008. Epub 2017 May 15. PMID: 29061412

Choo C et al (2014) Cluster analysis reveals risk factors for repeated suicide attempts in a multi-ethnic Asian population. *Asian J Psychiatr*. 2014 Apr;8:38-42. PMID:24655624

Ho RC et al (2014). Elderly suicide with and without a history of suicidal behavior: implications for suicide prevention and management. *Arch Suicide Res*. 2014;18(4):363-75. PMID:24828390

Choo CC et al (2018) Thematic Analysis of Medical Notes Offers Preliminary Insight into Precipitants for Asian Suicide Attempters: An Exploratory Study. *Int J Environ Res Public Health*. 2018 Apr 20;15(4). PMID: 29677141

Choo CC et al (2017b) Does ethnicity matter in risk and protective factors for suicide attempts and suicide lethality? *PLoS One*. 2017 Apr 20;12(4):e0175752. doi: 10.1371/journal.pone.0175752. eCollection 2017. PMID:28426687

[Response] Thank you so much for your comments and suggestions. This information is very important for our manuscript. Your comments have showed that you are very familiar with suicide studies, and we are so happy about your review on our manuscript. It is not only helpful for our manuscript, but also helpful for our knowledge about suicide in Asian.

In the introduction section, we mainly want to list some factors which may imply there were some suicide attempters who do not really want to die by suicide, and we listed mental disorder and impulsivity to explain it. Most of the factors you listed may be helpful to explain it, but there are also some factors which are not. (1) Religious belief and resolution of conflict are factors which are different between China and Western countries. In many studies including ours, the differences were explained by the culture differences 3 4. However, in our opinion, they are hard to explain there are some suicides who do not really want to die by suicide. (2) Factors that relationship issues, financial strain, socio-legal-academic, environmental stress, physical illness and pain are also associated suicide both in Asian and Western countries, but, in our opinion, they are also hard to explain there are some suicides who do not really want to die by suicide. (3) Mental disorder (including alcohol abuse) is a very important factor which is associated with suicide in Asian and Western countries, you also gave us so much information. The different percentages between Asian and Western countries can imply there may be some suicides do not really want to die by suicide, because mental disorder is a very important factor for suicide behavior. We have revised this part according to your suggestions. Thank you so much for your wonderful comments. Following is the revised section.

The percentage of mental disorder among suicide is different. In Western countries, there were about 90% of suicides who can be diagnosed with mental disorder 5. However, the percentage was lower in comparison with Asian countries. For example, they were only about 40- 70% in mainland China, 16.2% in Indian and 12.0% in Malaysia 6 7. Although mental disorder is also an important risk factor

for suicide behavior, other factors may also play roles in Asian suicides. Thus, exploring the characteristics of suicide attempters in China not only can help us to understand the differences between China and Western countries, but also can provide experiences for other Asian countries.

-3. On Pg 11, line 29, the authors stated that low level of impulsivity is associated with high level of suicide intent. The label of low level of impulsivity sounds contradictory. I suggest the authors to clarify the description of impulsivity.

[Response] Good catch! We have checked the description of impulsivity. The low level of impulsivity is associated with high level of suicide intent. This is really hard to understand. As in many suicide studies, impulsivity is a risk factor for suicide behavior. However, in the current study, the dependent variable was suicide intent, and we also can see it as the willing to die. The negative association means impulsivity is a protect factor for suicide intent. In other words, higher level of impulsivity was associated with low level of suicide intent. We also can see it as that suicide attempters with higher impulsivity do NOT want to die by suicide behavior. We wish our explanation can answer your comments. Thank you so much for your effort on our manuscript.

-4. On Pg 12, line 48, the authors stated that, "as we introduced above, many Chinese suicide attempters can be diagnosed with impulsivity suicide." First, impulsivity suicide is not a psychiatric diagnosis. Second, there is a possibility that the diagnosis of borderline personality disorder was missed as this condition is prevalence in women and characterised by impulsivity and recurrent self-harm behaviour. The authors should discuss the following finding:

In Asian patients, symptoms of borderline personality disorder were associated with risk of repeated overdoses with benzodiazepines and paracetamol (Choo et al 2014) and this condition is underdiagnosed.

Reference:

Choo C et al (2014) Cluster analysis reveals risk factors for repeated suicide attempts in a multi-ethnic Asian population. *Asian J Psychiatr.* 2014 Apr;8:38-42. PMID:24655624

[Response] Thank you so much for your nice comments. For the first questions, I do agree with your comments. Diagnose is not a good word here, we have revised it into "many Chinese suicide attempters can be categorized as impulsive 8."

For the second question, It is really a wonderful information for us. Borderline personality disorder characterized with impulsivity and self-harm behavior was not diagnosed in the current study. It is really a good point for us. We have cited this reference to meet this comment. The revised sentence is "One of the possible reasons may be that borderline personality disorder characterized with impulsivity and self-harm behavior was not diagnosed in the current study."

-5. On Pg 13, line 9, the authors mentioned about mental disorders without specifying the type of mental disorders. As more than half of suicide attempters were women and the mean age was around 31 years. The authors should discuss postnatal depression. There is a good BMJ Open article reviewing smartphone applications to engage women with postnatal depression. Smartphone applications may have the potential to engage women in rural area. Please refer to this study.

Zhang MW, et al (2017). Current status of postnatal depression smartphone applications available on application stores: an information quality analysis. *BMJ Open.* 2017 Nov 14;7(11):e015655. PMID:29138195

[Response] Thanks for your information. It is really a good paper which discussed the smart phone applications about postnatal depression. However, the main aim for this study was to determine the quality of the information content of phone application using validated scales. I think it is a little different from our aims in the current study.

In this study, we mainly talked about the characteristics of attempters who do not really want to die by suicide. There are following reasons why we do not want to discuss postnatal depression. Firstly, postnatal depression as one kind of depression had been diagnosed as depression in the current study, and we do not need to discuss it separately. Secondly, if we discuss the postnatal depression among females, we do not know how to explain the effect of mental disorder (or depression) on suicide among males. Thirdly, although more than half of suicide attempters were women and the mean age was around 31 years, the prevalence of postnatal depression was hard to be identified in

our study. Fourthly, although postnatal depression is a hot topic recently, it is not the topic in our study. But, I think it is really a good idea to explore the association between postnatal depression and suicide behavior in the future studies, and smart phone may play very important roles on mental disorders.

-6. Under discussion, the authors need to comment why there was no gender difference. This finding is different from other Asian suicide attempters living in societies with western influence. In other Asian studies, more males than females made attempts with high perceived lethality and high medical lethality (Choo et al 2017). The authors should discuss why the rural Chinese are different from other Asians.

Reference:

Choo CC et al (2017) Prediction of Lethality in Suicide Attempts: Gender Matters. *Omega (Westport)*. 2017 Jan 1;30222817725182. PMID:28828921

[Response] It is really a good catch! We have cited this reference in our manuscript, and discussed the gender problem with the following sentences in the revised manuscript.

In the current study, we did not find gender differences in suicide intent among suicide attempters. Previous studies demonstrated that male attempters tend to choose violent and lethal methods 9 , it was easy for us to conclude that males may experience higher levels of suicide intent. However, the choice of the suicide means may be caused by the higher level of violence for men compared to women, and we cannot conclude that men may have a higher intent to die. 10.

-7. Under discussion, I suggest the authors to develop a brief suicide assessment based on the factors identified and previous model of prediction. Please add the following statement in the discussion:

Based on our research findings, brief suicide risk assessment that include education level, negative life events and religious belief will achieve stronger predictive ability and reliability in suicide prediction based on the Affect-Behavior-Cognition model (Harris et al 2015).

Please include the following references:

Harris KM et al (2015) The ABC's of Suicide Risk Assessment: Applying a Tripartite Approach to Individual Evaluations. *PLoS One*. 2015 Jun 1;10(6):e0127442. PMID:26030590

[Response] Thanks for your references. Suicide risk assessment is a very important topic, and it is also very helpful for suicide prevention. According to your suggestions, we have cited this reference in our manuscript.

-8. I recommend the authors to discuss their findings with reference to the following important BMJ open articles in the discussion.

Xu H, et al (2016) A cross-sectional study on risk factors and their interactions with suicidal ideation among the elderly in rural communities of Hunan, China. *BMJ Open*. 2016 Apr 15;6(4):e010914. PMID: 27084285

Sun J et al (2015) Incidence and fatality of serious suicide attempts in a predominantly rural population in Shandong, China: a public health surveillance study. *BMJ Open*. 2015 Feb 11;5(2):e006762 PMID:25673439

Zhang M et al (2013).Pesticide poisoning in Zhejiang, China: a retrospective analysis of adult cases registration by occupational disease surveillance and reporting systems from 2006 to 2010. *BMJ Open*. 2013 Nov 21;3(11): e003510. PMID: 24270833

[Response] Thank you so much for your references. All of these publications have been cited in our manuscript.

Reviewer: 2

Reviewer Name: Maurizio Pompili

Institution and Country: Sapienza University of Rome

Please state any competing interests or state 'None declared': None

-The authors reported original research report pointing to features for medically serious suicide attempters whose lethality seem limited. The paper is of interest for the journal. I have however some comments to report.

[Response] Thank you so much for your insightful comments on our manuscript. We have studied

these comments carefully and tried our best to revise the manuscript in a more reasonable way. We are hopeful our responses meet your kind consideration for the acceptance of this paper.

First, aims and hypothesis should be better described.

[Response] Thanks for your suggestions. We have revised the aims and hypothesis in our manuscript. Following are our revised paragraphs in the abstract and introduction section.

Many previous studies implied that there were many Chinese suicide attempters who did not want to die by suicide. In the current study, we aim to explore the factors which are associated with the low level of suicide intent. In other words, we can also learn the features for medically serious suicide attempters who do not have a strong intent to die in rural china.

Thus, in the current study, we explored factors associated with low level of suicide intent. It was helpful for us to understand the features for medically serious suicide attempters who do not have a strong intent to die in rural China, which may inform intervention and prevention strategies for at-risk individuals in rural China.

Second, authors should clearly describe what a suicide attempt it, and possibly refer to the revision of nomenclature in suicidology such as Silverman et al, 2007 (two papers).

[Response] Thank you so much for your references. The two articles are so important for the nomenclature about suicide. We have read these two papers carefully, and cite them to describe what suicide attempt is in our study. As the description in the references, there are some overlaps between medically serious suicide attempt and suicide. It was also identified in previous studies, and we think medically serious suicide attempt (also named nearly lethal suicide attempt) may be also a good word to use for our sample.

Third, authors may wish to discuss the concept of lethality as the probability to die by suicide and refer to the thought of Shneidman. Authors may also need to add some details referred to the enrollment of patients, who actually assessed them, if more than a psychiatrist was involved.

[Response] Thanks for your reminder. It is really a good catch! Lethality is a very important factor which is associated with the results of suicide attempt. Many previous studies have proved this finding. In our opinion, lethality is different from suicide intent, although they should highly correlate. For lethality, it is a relatively objective concept. It mainly depends on the suicide means. Suicide intent is a relatively subjective concept, and it is about the willing to die. As you know, many items in SIS are subjective, such as the aims, the attitude, and so on. I do agree with you, it is important for the discussion about lethality. However, we worry about that this would get some misunderstanding about the results for our study. As a response, we have listed it in the limitation section to explain it.

Following is the revised sentence.

Finally, lethality, which is an important factor associated with suicide intent, was not investigated in our study.

For the enrollment of patients, we were informed by all of departments of hospital emergency in each rural county, and we would be their home to make sure the suicide behaviors according the definition introduced in the manuscript before the interview. Following is the revised sentence.

The enrollment of patients followed the definition of suicide attempt and deemed medically serious. The interview was conducted by the master and Ph.D. students in each province. They are medical student in each university. The diagnosis of mental disorder was made by the psychiatrists with the written information obtained by the trained interviewers for each suicide attempt. Following is the revised sentence.

All of the interviewers received training about the study and were master or PhD level students in the medical school.

There were more than one psychiatrist was involved. In each province, there was one psychiatrist employed for this study. In the whole study, there were two psychiatrists who were involved. Following is the revised sentence.

There was one psychiatrist to make the diagnosis in each province.

-Finally, I would point to the prevention of such act and discuss the following paper: The communication of suicidal intentions: a meta-analysis. Psychol Med. 2016

[Response] Thank you so much for your suggestion. It is really a wonderful work about suicide

communication. We have learned this publication carefully, and cited it in our manuscript.

Reference:

1. Qin P, Mortensen PB. Specific characteristics of suicide in China. *Acta Psychiatr Scand* 2001;103(2):117-21.
2. Hvistendahl M. Making Sense of a Senseless Act. *Science* 2012;338(6110):1025-27. doi: 10.1126/science.338.6110.1025
3. Zhang J, Xu HL. The effects of religion, superstition, and perceived gender inequality on the degree of suicide intent: A study of serious attempters in China. *Omega: J Death Dying* 2007;55(3):185-97. doi: 10.2190/OM.55.3.b
4. Wang Z, Koenig Harold G, Ma W, et al. Religious involvement, suicidal ideation and behavior in mainland China. *Int J Psychiatry Med* 2015;48(4):299-316. doi: 10.2190/PM.48.4.e
5. Cavanagh JT, Carson AJ, Sharpe M, et al. Psychological autopsy studies of suicide: a systematic review. *Psychol Med* 2003;33(3):395-405.
6. Zhang J, Xiao S, Zhou L. Mental disorders and suicide among young rural Chinese: a case-control psychological autopsy study. *Am J Psychiatry* 2010;167(7):773-81. doi: 10.1176/appi.ajp.2010.09101476 [published Online First: 2010/04/17]
7. Choo CC, Harris KM, Chew PKH, et al. Does ethnicity matter in risk and protective factors for suicide attempts and suicide lethality? *PloS one* 2017;12(4):e0175752. doi: 10.1371/journal.pone.0175752
8. Li X, Phillips M, Wang Y, et al. The comparison of impulsive and non-impulsive attempted suicide. *Chinese Journal of Nervous and Mental Diseases* 2003;29(1):27-31.
9. Choo CC, Harris KM, Ho RC. Prediction of Lethality in Suicide Attempts: Gender Matters. *Omega (Westport)* 2017;30222817725182. doi: 10.1177/0030222817725182
10. Sun L, Zhang J. Characteristics of Chinese rural young suicides: Who did not have a strong intent to die. *Compr Psychiatry* 2015;57(1):73-78. doi: 10.1016/j.comppsy.2014.11.019

STROBE Statement—checklist of items that should be included in reports of observational studies
Item No Recommendation Line number

Title and abstract 1 (a) Indicate the study's design with a commonly used term in the title or the abstract 1-2;

(b) Provide in the abstract an informative and balanced summary of what was done and what was found 22-37;

Introduction

Background/rationale 2 Explain the scientific background and rationale for the investigation being reported 86-93;

Objectives 3 State specific objectives, including any prespecified hypotheses 94-95;

Methods

Study design 4 Present key elements of study design early in the paper 101-103;

Setting 5 Describe the setting, locations, and relevant dates, including periods of recruitment, exposure, follow-up, and data collection 101-114;

Participants 6 (a) Cohort study—Give the eligibility criteria, and the sources and methods of selection of participants. Describe methods of follow-up

Case-control study—Give the eligibility criteria, and the sources and methods of case ascertainment and control selection. Give the rationale for the choice of cases and controls

Cross-sectional study—Give the eligibility criteria, and the sources and methods of selection of participants 108-114;

(b) Cohort study—For matched studies, give matching criteria and number of exposed and unexposed

Case-control study—For matched studies, give matching criteria and the number of controls per case

--

Variables 7 Clearly define all outcomes, exposures, predictors, potential confounders, and effect modifiers. Give diagnostic criteria, if applicable 136-198;

Data sources/ measurement 8* For each variable of interest, give sources of data and details of methods of assessment (measurement). Describe comparability of assessment methods if there is more than one group 136-198;

Bias 9 Describe any efforts to address potential sources of bias 284-291;

Study size 10 Explain how the study size was arrived at ---

Quantitative variables 11 Explain how quantitative variables were handled in the analyses. If applicable, describe which groupings were chosen and why 200-204;

Statistical methods 12 (a) Describe all statistical methods, including those used to control for confounding 200-204;

(b) Describe any methods used to examine subgroups and interactions 200-204;

(c) Explain how missing data were addressed

(d) Cohort study—If applicable, explain how loss to follow-up was addressed

Case-control study—If applicable, explain how matching of cases and controls was addressed

Cross-sectional study—If applicable, describe analytical methods taking account of sampling strategy 200-204;

(e) Describe any sensitivity analyses --

Results Line number

Participants 13* (a) Report numbers of individuals at each stage of study—eg numbers potentially eligible, examined for eligibility, confirmed eligible, included in the study, completing follow-up, and analysed 26

(b) Give reasons for non-participation at each stage --

(c) Consider use of a flow diagram --

Descriptive data 14* (a) Give characteristics of study participants (eg demographic, clinical, social) and information on exposures and potential confounders 206-213

(b) Indicate number of participants with missing data for each variable of interest --

(c) Cohort study—Summarise follow-up time (eg, average and total amount) --

Outcome data 15* Cohort study—Report numbers of outcome events or summary measures over time --

Case-control study—Report numbers in each exposure category, or summary measures of exposure --

Cross-sectional study—Report numbers of outcome events or summary measures 220-221

Main results 16 (a) Give unadjusted estimates and, if applicable, confounder-adjusted estimates and their precision (eg, 95% confidence interval). Make clear which confounders were adjusted for and why they were included 220-214

(b) Report category boundaries when continuous variables were categorized --

(c) If relevant, consider translating estimates of relative risk into absolute risk for a meaningful time period --

Other analyses 17 Report other analyses done—eg analyses of subgroups and interactions, and sensitivity analyses --

Discussion

Key results 18 Summarise key results with reference to study objectives 226-231

Limitations 19 Discuss limitations of the study, taking into account sources of potential bias or imprecision. Discuss both direction and magnitude of any potential bias 284-291

Interpretation 20 Give a cautious overall interpretation of results considering objectives, limitations, multiplicity of analyses, results from similar studies, and other relevant evidence 284-291

Generalisability 21 Discuss the generalisability (external validity) of the study results 296-297

Other information

Funding 22 Give the source of funding and the role of the funders for the present study and, if applicable, for the original study on which the present article is based 304-306

*Give information separately for cases and controls in case-control studies and, if applicable, for

exposed and unexposed groups in cohort and cross-sectional studies.

Note: An Explanation and Elaboration article discusses each checklist item and gives methodological background and published examples of transparent reporting. The STROBE checklist is best used in conjunction with this article (freely available on the Web sites of PLoS Medicine at <http://www.plosmedicine.org/>, Annals of Internal Medicine at <http://www.annals.org/>, and Epidemiology at <http://www.epidem.com/>). Information on the STROBE Initiative is available at www.strobe-statement.org.

VERSION 2 – REVIEW

REVIEWER	Roger Ho Department of Psychological Medicine Yong Loo Lin School of Medicine National University of Singapore, Singapore, pcmrhcm@nus.edu.sg
REVIEW RETURNED	17-Aug-2018
GENERAL COMMENTS	I recommend publication.
REVIEWER	Maurizio Pompili Sapienza University of Rome
REVIEW RETURNED	16-Aug-2018
GENERAL COMMENTS	The authors addressed my comments and the paper appears suitable for possible publication in the journal